# Influence of Line Processing Parameters on Properties of Carbon Fibre Epoxy Towpreg

**Murat Çelik, Thomas Noble, Frank Jorge, Rongqing Jian, Conchúr M. Ó Brádaigh** and **Colin Robert** *

School of Engineering, Institute for Materials and Processes, The University of Edinburgh,
Edinburgh EH9 3FB, UK; m.celik@ed.ac.uk (M.Ç.); tnoble2@exseed.ed.ac.uk (T.N.); f.a.d.jorge@sms.ed.ac.uk (F.J.);
r.jian@sms.ed.ac.uk (R.J.); c.obradaigh@ed.ac.uk (C.M.Ó.B.)
* Correspondence: colin.robert@ed.ac.uk

**Abstract:** This paper explores the performance of low-cost unidirectional carbon fibre towpregs with respect to line production speed and fibre volume fraction. Using an automated production line, towpregs were produced at different production speeds, resulting in modified fibre volume fractions. The towpregs were used to manufacture unidirectional composite plates, which were then tested to evaluate mechanical performance. The fibre straightness and interfacial void ratio of the composite plates were determined by statistical analysis of the samples' optical micrographs. The results demonstrate that adjusting the line production speed enables targeted fibre volume fractions (FVF) to be reached, resulting in the composites having different mechanical performances (2039 MPa and 2186.7 MPa tensile strength, 1.26 and 1.21 GPa flexural strength for 59.8% and 64.4% FVF, respectively). It was shown that at lower production speeds and FVF, composites exhibit good consolidation and low porosity, which is highlighted by the better interlaminar shear strength performances (8.95% increase), indicating the limitations of manufacturing very high FVF composites. Furthermore, it was concluded that fibre straightness plays a key role in mechanical performance, as samples with a lesser degree of fibre straightness showed a divergence from theoretical tensile properties.

**Keywords:** fibre volume fraction; towpreg; advanced composite manufacturing; powder epoxy

## 1. Introduction

Out-of-autoclave (OOA) towpreg epoxy composites are ideal candidates to provide adequate strength and stiffness for large composite structures, such as tidal/wind turbine blades and marine or automotive applications, while being cost-effective due to the scalability and compatibility with inexpensive mould heating systems [1]. In recent years, the need for low-cost, high-performance OOA prepreg composites has led to an interest in powder-epoxy systems, which are solid at room temperature, melt between 40 and 60 °C, and cure at 180 °C. It has been demonstrated that powder–epoxy systems can be used to produce vacuum-bag-only (VBO) prepregs, and that they exhibit comparable mechanical performance to conventional epoxy composites [2,3]. Furthermore, low viscosity and low curing exotherms of powder–epoxies make them attractive for thick section composites [2–4]. Owing to their low viscosity, a complete epoxy wet-out within the mould is possible with powder–epoxy systems under vacuum only, which results in superior strength and fracture toughness due to the strong interfacial bonding between the matrix and the fibres [5]. Furthermore, the powder can be stored at room temperature thanks to its thermal stability, and little to no volatile organic compounds (VOC) are released during the production [6].

In this context, a novel powder–epoxy-based pilot towpregging line has been developed [7] to manufacture unidirectional carbon fibre towpregs (or tapes) that are compatible with advanced composite manufacturing methods such as Automated Fibre Placement

(AFP), pultrusion, or filament winding. The main goal of this system is to allow for low-cost, high-quality, and high-speed manufacturing of unidirectional carbon fibre-reinforced polymer (UD-CFRP) towpregs. Automation of the reported system [7] is in progress, which entails better process control and higher production speeds, therefore lowering the cost of the overall composite production. On one hand, higher manufacturing speeds increase the fibre volume fraction (FVF) of the composite and alter the mechanical properties remarkably. Fibres display better mechanical properties than resin; thus, a higher FVF allows for higher strength and stiffness due to a higher fibre-to-resin ratio. On the other hand, too high FVF results in a lack of wetting for individual fibres, macroscopically leading to early interfacial failure due to poor fibre–matrix interface bonding and energy transfer ability [8]. As such, it was deduced that the optimal FVF for UD-CFRP plates in terms of the ultimate tensile strength (UTS) is 56–60%, as UTS decreases for higher FVFs due to the lack of completely wet-out fibres [9]. Since there is no additional compaction pressure as in the autoclave process, reaching the desired FVF values with OOA systems is not an easy task. Courter et al. [10] demonstrated that significantly higher FVF values could be obtained by autoclave curing when compared to oven curing for the same type of prepreg materials. Still, up to 60% FVF can be achieved with VBO prepregs [11]; however, the out-life of the prepregs plays a key role in the composite production phase, and a few weeks of out-life substantially decreases the processibility of the prepregs [12] due to partial curing increasing the minimal pre-gel viscosity, therefore reducing the wetting ability.

The powder–epoxy system used in this study is solid and stable at ambient temperature and will not start to cure before 145 °C, due to its heat-activated curing mechanism [7]. Therefore, it has excellent storage performance at room temperature; moreover, a consistent FVF can be obtained by carefully adjusting the towpreg line process parameters such as speed, tension, or temperature. Most of the standard VBO prepreg resins are highly reactive and cannot maintain low viscosities for a long time [11], whereas powder–epoxy viscosity remains very low at lower temperatures [4], and complete wet-out of the fibre bed can be achieved with vacuum only—without the need for additional compaction pressure. Controlling the FVF while maintaining a high production speed in the towpregging line would allow the manufacture of high-performance composites, such as wind or tidal turbines, at a low cost. In this study, a consistent towpreg FVF was maintained for a certain production speed in a powder–epoxy based towpregging line. Two production speeds, 3 and 5 m/min, were used to produce towpreg to investigate the FVF/production speed relationship and its influence on the overall mechanical performances, while all other processing parameters, such as tension and temperature, were kept constant and monitored.

## 2. Experimental Section

### 2.1. Materials

Powder epoxy (PE6405, 1220 kg/m$^3$) supplied by FreiLacke and designed by Swiss CMT AG was used to manufacture the towpreg and then the CFRP plates. One of the main advantages of the powder–epoxy is that it starts to melt at 40–60 °C, reaches the minimum viscosity at 120 °C, but does not cure until 145 °C [4]. This feature provides versatility in the production phase by allowing the potential of separating the impregnation and curing of the composite parts. In addition, it is possible to co-cure different composite parts using powder–epoxy [13] instead of using adhesives [7]. Finally, as the powder epoxy is solid and stable at room temperature, storage costs are substantially lower than standard prepreg systems.

For the carbon fibre tows, commercially available Toray T700S-24K-50C (1% sizing agent) was used, as it showed the best performances comparatively to other fibre systems [14].

### 2.2. Towpreg Production

The powder–epoxy towpregging pilot line was originally developed by Robert et al. [7]. Since then, a second more instrumented tapeline has been developed in the framework of a

CIMComp Hub fellowship project to automate the production process. New sensors were added, including OS-PC30-2M-1V infrared temperature sensors (OMEGA Engineering), an SFD 500T tension sensor (Hans Schmidt & Co GmbH, Waldkraiburg, Germany), and a P500 rotary sensor (Positek, Cheltenham, UK). Control hardware included a B35-HM magnetic particle brake (Placid Industries Inc., Elmira, NY, USA) and EM-175 DC motor controllers. Automated temperature and tension control was achieved using software-side PIDs on LabVIEW (National Instruments Corp., Austin, TX, USA). The towpreg production in the line can be summarised in the following steps, as shown in Figure 1:

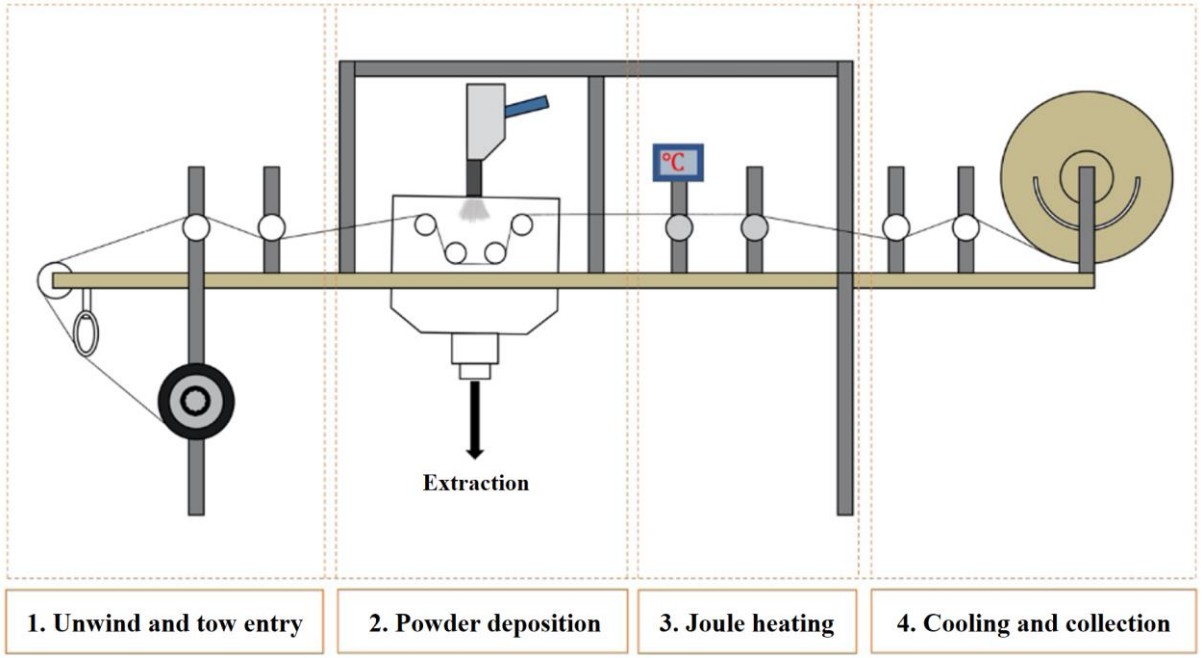

**Figure 1.** Diagram of the tapeline system.

- Carbon fibre tows are unwound from the reel, and the tension is maintained by a magnetic brake that automatically adjusts the tension based on the data from the tension sensor.
- Powder–epoxy is electrostatically charged and sprayed on the carbon fibre tow as it moves.
- An electrical current is supplied via a power controller between two conductive metal rollers to initiate powder melt by the Joule effect [15]. The temperature can be adjusted according to data from an infrared sensor.
- Cooled towpreg is collected on a drum after it passes through a series of rollers.

Automation of the pilot towpregging line is necessary for increasing the productivity and accuracy of the production while lowering costs. Additionally, the towpreg quality can be maintained for high production volumes as the entire system is being monitored with the data obtained from the sensors. In order to control and monitor the system, a human–machine interface (HMI) has been created using LabVIEW software, which allows the user to observe and record key process parameters, such as tension, temperature, or speed. All the parameters can be controlled either manually or by PID controllers via the HMI.

Reliability and consistency were the goals for towpreg production, which are easier to achieve at slower production speeds. Therefore, the towpreg was produced at two different speeds, 3 and 5 m/min, while keeping all other parameters constant (temperature, tension, etc.). The produced towpreg was cut into equal-sized so-called "strips" (55 cm length),

and the FVF of the strip samples ($FVF_{strip}$) was calculated for each production run using Equation (1) [16]:

$$FVF = \frac{\rho_m}{\rho_m + \rho_f \left( \frac{1}{FWF} - 1 \right)} \tag{1}$$

where $\rho_m$ is the matrix density, $\rho_f$ is the fibre density, and $FWF$ is the fibre weight fraction, which are determined by a precision scale.

### 2.3. Composite Plate Production

For further mechanical characterisation tests, unidirectional carbon fibre composite (UD-CFRP) plates were manufactured from the produced powder epoxy towpreg strips. Composite plate production consisted of the following stages, as shown in Figure 2:

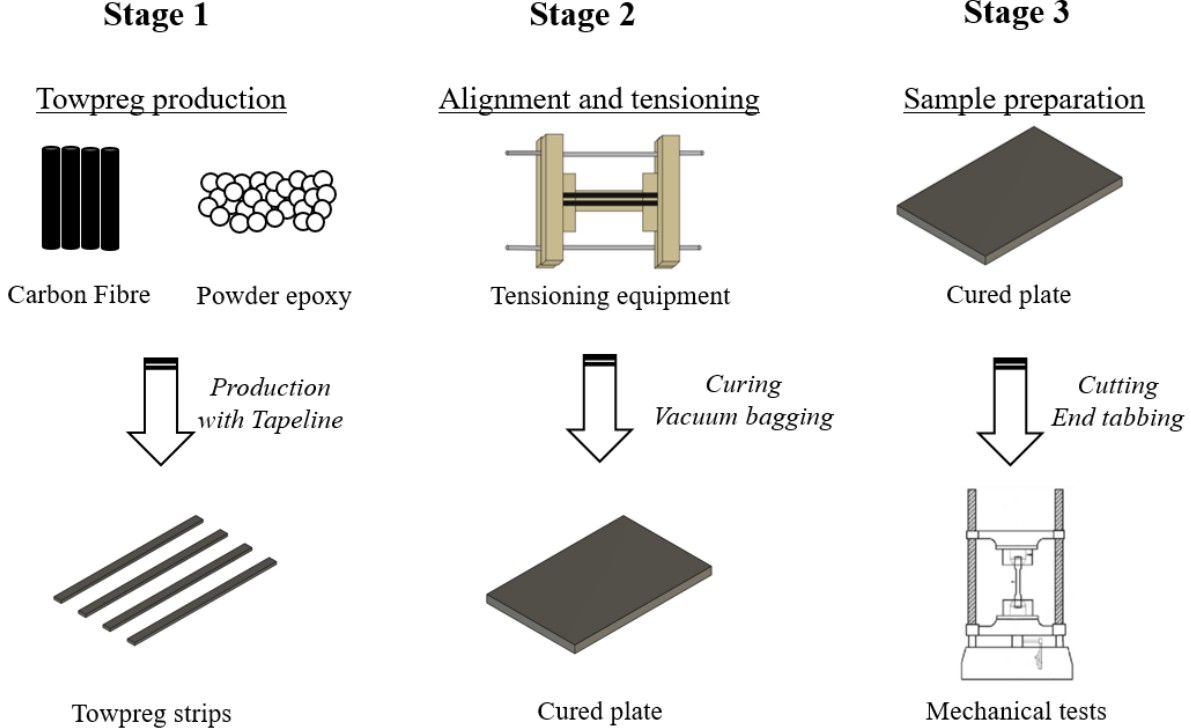

**Figure 2.** Different stages of the UD composite plate production.

Stage 1: Towpreg was produced with the tapeline system at different production speeds; then, it was cut into strips.

Stage 2: Towpreg strips were aligned by custom tensioning equipment [14]. Then, the tensioned preform was vacuum-bagged and cured to manufacture unidirectional composite plates, with 5 layers of towpreg strips being used to produce 1 mm thick composite plates. The thermal cycle for curing consisted of a drying stage dwell at 35 °C for 5 h, which was followed by an isothermal dwell at 120 °C for 2 h for the sintering and melting stage and another isothermal dwell at 180 °C for 2 h to complete curing. One hour ramping time was used between all dwells.

Stage 3: To prepare samples for mechanical characterisation, UD-CFRP composite plates were cut using a wet saw to ensure smooth edges. Then, mechanical tests were carried out for performance characterisation.

After the production of cured plates and extraction of samples, UD-CFRP samples were weighed again, and the FVF of the composite plates ($FVF_{plate}$) was calculated using Equation (1). Peel ply was used as a separation layer between the preform and tensioning equipment (Figure 3a). The excess resin was absorbed by the peel ply during the curing stage (Figure 3c), which resulted in an FVF increase compared to the initial FVF value ($FVF_{strip}$). Table 1 summarises the FVF of the towpreg samples and composite plates at

different process parameters including speed, tension, temperature, and electrostatic spray gun settings. As can be seen from the table, both the $FVF_{strip}$ and $FVF_{plate}$ increase with increasing production speed, since the duration that the carbon fibre tows spend within the electrostatic spraying chamber becomes shorter for higher speeds, resulting in less powder deposited in the towpreg. For the electrostatic powder deposition, the electrostatic gun capacity was set to 40% for all experiment sets.

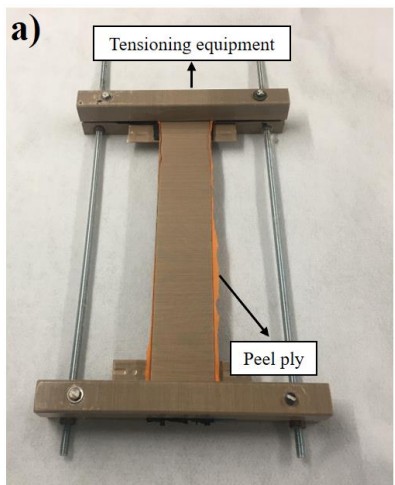 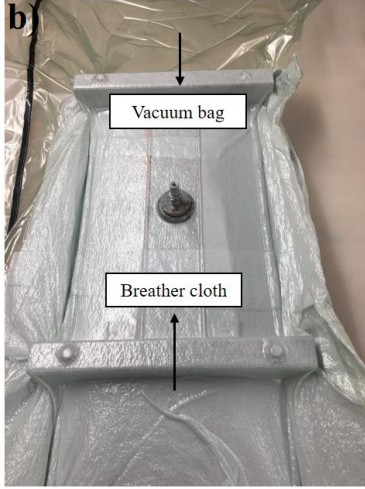 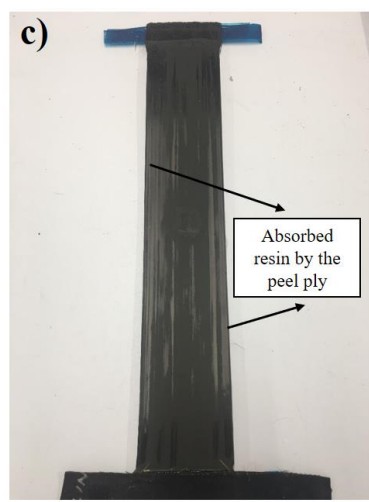

**Figure 3.** (**a**) Towpregs in the tensioning equipment (enclosed within the mould and not visible) and peel ply (orange). (**b**) Vacuum bagging of the tensioning equipment. (**c**) Cured composite plate and visible spots from absorbed resin by the peel ply (sample set 2 case).

**Table 1.** FVF of the towpreg strips and composite plates at different processing parameters.

|  | Line Speed (m/min) | Line Tension (N) | Heating Temperature (°C) | Gun Flow Air (%) | Total Flow (Air + Powder) (%) | Strip FVF | Plate FVF |
|---|---|---|---|---|---|---|---|
| Sample set 1 | 3 | 20 | 120 | 99 | 40 | 53.1% | 56.5% |
| Sample set 2 | 5 | 20 | 120 | 99 | 40 | 62.5% | 64.2% |

### 2.4. Tensile Tests

Tensile tests were carried out on 190 × 15 mm (length and thickness, respectively) samples according to BS EN ISO 527-5 standard. Glass–epoxy laminates were bonded to the samples for end tabbing. For each set, 5 samples were tested in the fibre (0°) direction using an MTS Criterion model 45 (C45.305) electromechanical universal test system at a constant crosshead speed of 2 mm/min. A digital image correlation (DIC) module (Imetrum Advantage video extensometer) was used for the 2D strain measurement of the samples. The samples were collected after the test to analyse the fracture surfaces via Scanning Electron Microscopy (SEM).

### 2.5. Flexural Tests

Flexural performance of the 90 mm × 15 mm samples was characterised by four-point bending tests according to BS EN ISO 14125. Twelve samples (6 for each speed) were tested with Instron Model 3369 Dual column Tabletop Test system. The test fixture had an 81 mm outer span and a 27 mm inner span. The DIC module (Imetrum Advantage video extensometer) captured the deflection, and the crosshead speed was 2 mm/min.

### 2.6. Interlaminar Shear Strength

The interlaminar shear strength (ILSS) of the UD-CFRPs plates was evaluated according to BS EN ISO 14130:1998 on twelve 20 mm × 13 mm sized samples. The same Instron

test machine was used for ILSS tests with a crosshead speed of 1 mm/min. The load cell capacity for the system was 50 kN.

### 2.7. Scanning Electron Microscopy (SEM)

The surface topography of the samples and fracture cross-sections were analysed by a Hitachi TM4000 Plus Scanning Electron Microscope (SEM) in Back Scattered Electrons (BSE) mode with an acceleration voltage of 15 kV under vacuum conditions.

### 2.8. Optical Microscopy

The fibre distribution, FVF, and porosity of the UD-CFRP samples were analysed by optical microscopy. Before the analysis, the samples were potted using liquid epoxy resin and non-stick moulds; then, they were polished by a surface grinder. The polishing procedure included sanding with varying grit sizes (P800, P1200, and P1600) followed by polishing by diamond-based dispersion (5, 3, and 1 micron). Then, the samples were analysed by a Zeiss Axioskop 2 MAT model microscope that is connected to a computer via an AxioCam MRc 5 camera. Fifty images were taken for each sample set, aiming to eliminate errors from a low sample size. The images were post-processed with ImageJ software. Firstly, images were converted to black and white to create a better contrast, as shown in Figure 4; then, the threshold for the pixels was set accordingly to capture fibres, matrixes, and voids in the sample. Then, the area of each component was calculated by ImageJ to validate the FVF values calculated with Equation (1). The void fraction and fibre circularity was also calculated with ImageJ, using the micrographs.

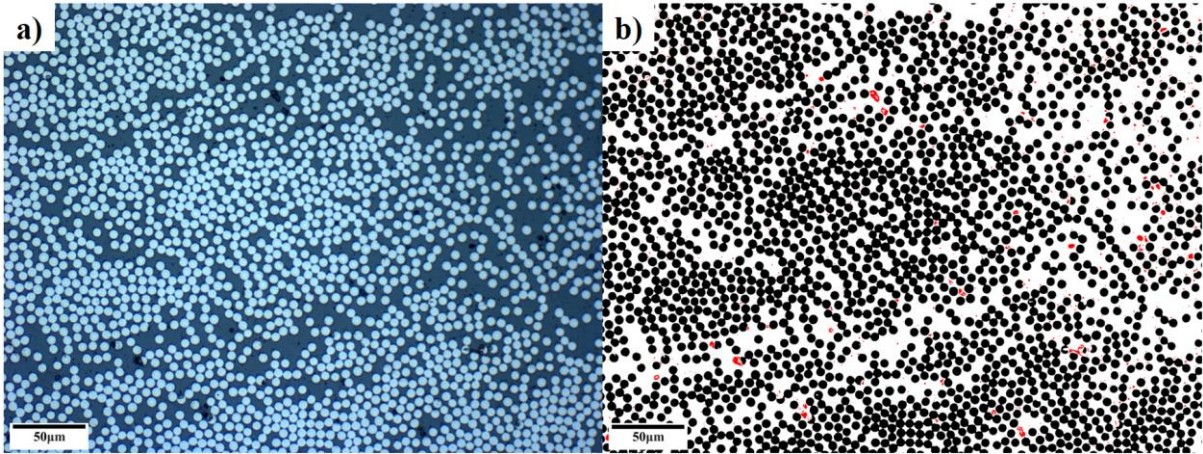

**Figure 4.** (**a**) Optical microscopy of the UD-CFRP sample cross-section for 59.8% FVF. (**b**) Resin (white), fibres (black), and voids (red) identified by ImageJ software filters.

### 2.9. Interfacial Void Content Analysis

Voids are one of the main manufacturing defects in composites, influencing the mechanical properties considerably by acting as failure initiation points [17]. The tensile, compressive, flexural, and interlaminar shear strength of the composites are adversely affected by the voids [18]; moreover, fatigue life deteriorates seriously [19]. The spatial distribution of the voids is a key factor for the performance of the composite. In the presence of voids at the fibre–matrix interface, adhesion between the fibre and matrix is weakened, and load transfer is impaired [20]. As it is dominated by the properties of matrix and matrix–fibre interface [19], interlaminar shear strength is susceptible to the interfacial voids in particular. For the samples that were produced at different production speeds, interfacial void content was determined using ImageJ and a custom LabVIEW algorithm. Using the samples analysed by the Zeiss Axioskop 2 MAT model microscope, ImageJ was used to find geometric information on the fibres, matrix, and voids. Data for all samples was exported to a csv file. This included information on the locations of the fibres and

voids, the maximum diameter of each fibre (using a maximum calliper rule), and the area of the fibres and voids.

An algorithm was applied to the data from each sample image, which isolated interfacial voids from bulk voids. This was achieved using the following methodology.

- The coordinates of a fibre were compared to the locations of the voids for a sample using Equation (2).

$$\sqrt{\left(x_f - x_v\right)^2 + \left(y_f - y_v\right)^2} = d \tag{2}$$

where:
$x_f$          The x-coordinate of the fibre centre point
$x_v$          The x-coordinate of the void centre point
$y_f$          The y-coordinate of the fibre centre point
$y_v$          The y-coordinate of the void centre point
$d$    Distance between the centre points of the fibre and void

- Voids that were outside the fibre radius $r_f$ and within a set distance of the outer edge of the fibre $r_{f+a}$ were recorded using comparator Equation (3), along with their respective area (Figure 5). These were classified as interfacial voids.

$$if\ d > r_f\ and\ d < r_{f+a} \tag{3}$$

where:
$r_f$          The radius of the fibre using a maximum calliper
$r_{f+a}$    The radius of the fibre plus a set distance (0.25, 0.5, and 1 μm)

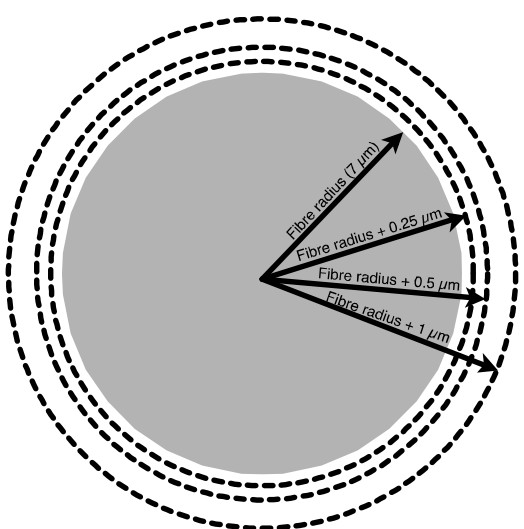

**Figure 5.** Boundary regions used to search for interfacial voids.

- This process was repeated for every fibre in a sample picture, and the individual areas were summed to provide a total area of interfacial voids (Equation (4)).

$$\sum A_{n\ interfacial\ voids} = A_1 + A_2 + \ldots + A_n \tag{4}$$

- Then, the interfacial void area was compared with the total void area found previously using optical microscopy to find the percentage of interfacial voids compared to the total void fraction.

$$\frac{A_{interfacial\ voids}}{A_{voids}} \times 100 = \%\ of\ interfacial\ voids\ to\ bulk\ voids \tag{5}$$

- This process was repeated for the other sample pictures, and an average was taken of the percentage of interfacial voids at the two towpreg line speeds, 3 and 5 m/min. Figure 6 illustrates the locations of interfacial and bulk voids that were captured by the algorithm (circles sizes are not representing the void areas).

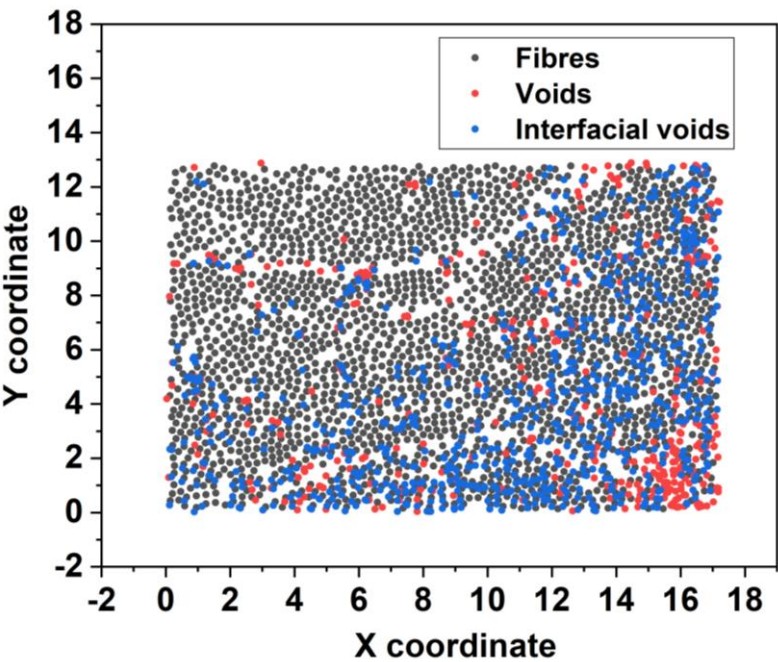

**Figure 6.** Bulk and interfacial voids detected by the algorithm.

## 3. Results and Discussion

### 3.1. Fibre Volume Fractions: Strip, Plate, and Bulk

Two different production line speeds (3 and 5 m/min) were used in this study. As expected, $FVF_{strip}$ for the 3 m/min samples was lower, 53.1%, since the amount of powder supplied decreases with increasing production speed, as the towpreg is coated with epoxy powder for less time. An $FVF_{strip}$ value of 62.5% was obtained with the 5 m/min samples, which indicates that the resin may not have completely saturated the fibres. After the curing stage, fibre volume fractions increased to 56.5% and 64.2% for 3 m/min and 5 m/min samples, respectively, which were named $FVF_{plate}$. This increment in FVF, as mentioned above, was caused by the fact that the peel ply absorbed some fraction of the resin from the tensioned preform during curing. Then, the calculated values of $FVF_{plate}$ from Equation (1) were compared with the post-processing of optical microscopy images using ImageJ software. The FVF values obtained by the optical microscopy images are presented in Table 2. FVF values calculated by the optical microscopy for the 5 m/min samples matched very well with $FVF_{plate}$, which were estimated from Equation (1), whereas a deviation was observed for 3 m/min samples. Due to the higher resin content of the 3 m/min samples, resin-rich zones were found around sample boundaries (edges). During the optical microscopy analysis, boundaries of the samples were not considered; thus, an increment of the FVF was observed for 3 m/min samples, since the resin-rich zones were excluded. Having less resin content, 5 m/min samples did not exhibit such resin-rich zones; hence, the FVF value calculated by the optical microscopy did not differ significantly from the previous FVF values. Then, the FVF values obtained by the optical microscopy were regarded as the best representation of the bulk composite fibre volume fraction and named as $FVF_{Optical}$.

**Table 2.** FVF values of the composite plates, estimated by optical microscopy.

| Line Speed (m/min) | FVFstrip (%) | FVFplate (%) | FVFOptical (%) | Porosity (%) |
|---|---|---|---|---|
| 3 | 53.1% | 56.5% | $59.8 \pm 4.04$ | $0.62 \pm 0.09$ |
| 5 | 62.5% | 64.2% | $64.4 \pm 2.20$ | $0.82 \pm 0.19$ |

As mentioned before, voids in a composite material are detrimental to its performance; ILSS has been shown to decrease by 6% per 1% of void (up to 4% void fraction) [21]. Optical microscopy was used to investigate the porosity of the UD-CFRP composite laminate samples; in addition, the FVF and fibre distribution were also analysed. Post-processing was conducted using ImageJ software. When compared with other conventional and commercial methods [19], the void fraction of the samples is relatively low (<1%), and it can be inferred that consolidation is excellent in the manufactured plates.

### 3.2. Interfacial Voids

Results for the percentage of interfacial voids as a fraction of the total voids are shown below in Figure 7. Regardless of the search criterion (distance outside fibre radius), generally, a higher interfacial void content was found for samples produced at the higher speed of 5 m/min. The ratios of interfacial voids were 3.21% and 6.07% at 0.25 μm, 15.9% and 22.3% at 0.5 μm, 53.01% and 56.29% at 1 μm distance outside fibre radius at 3 m/min and 5 m/min production speed, respectively. The higher interfacial void ratio at the higher production speed (i.e., higher FVF) was attributed to the lesser resin content within the towpreg compared to 3 m/min production speed samples.

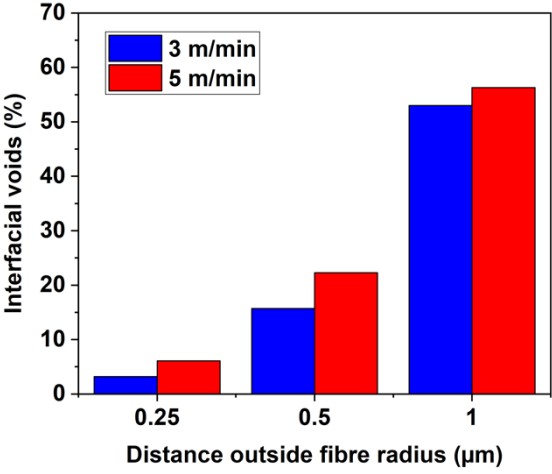

**Figure 7.** Percentage of interfacial voids as a fraction of the total voids.

### 3.3. Fibre Straightness

Mamalis et al. [14] found that the fibre straightness of unidirectional carbon fibre tows had a significant impact on the mechanical performance of the composite. Hence, ImageJ software was used to analyse the degree of fibre straightness in the samples. The minimum ($D_{i,min}$) and maximum diameter ($D_{i,max}$) of each fibre in the optical micrographs were measured, which were used to calculate the ratio $D_{i,total} = D_{i,min}/D_{i/max}$. Then, the following equation [14] can be used to obtain the direction parameter *f*:

$$f = \frac{\sum_{i=1}^{n} D_{i,total}}{n} \tag{6}$$

where *n* is the number of carbon fibre filaments investigated for each micrograph. The direction parameter *f* indicates how far a fibre deviates from a perfect circle and can be used as a measure of fibre straightness. A total of 75 micrographs were analysed, each containing

≈2000 visible fibres. The number of fibres analysed (*n*) was greater than 150,000, therefore providing strong analytical results. The estimated direction parameter *f* for each FVF is shown in Figure 8. For both sample sets, a similar value of *f* (≈0.91) was obtained, which was influenced by the tensioning conditions during curing. For similar powder–epoxy towpreg and the same fibre type (T700S-24K-50C), an *f* value of 0.95 was reported for a tensioned system [14]. A lower degree of fibre straightness points to a slight misalignment of fibres caused by the tensioning apparatus, resulting in lower mechanical performances comparatively [14].

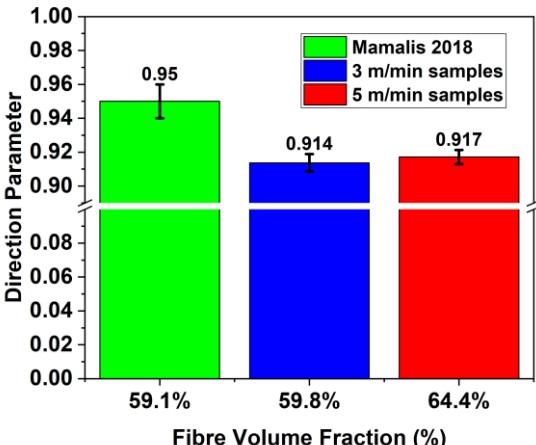

**Figure 8.** Direction parameter for different FVF samples compared to that of Mamalis et al. [14].

### 3.4. Tensile Performance

Different production speeds of the towpregging line resulted in different mechanical and fractographic behaviour. The average tensile strength of 59.8% and 64.4% FVF samples were 2039 MPa and 2187 MPa, whereas the average modulus values were 116 GPa and 138 GPa, respectively. As illustrated in Figure 9a, both strength and stiffness increased by 7.24% and 9.28%, respectively, for the higher FVF material, or in other words, with the production speed. Moreover, 64.4% FVF samples exhibited slightly larger standard deviation in modulus (±16.3 GPa compared to ±9.79 GPa), which is believed to be a result of lower homogeneity across the samples due to less consistent consolidation.

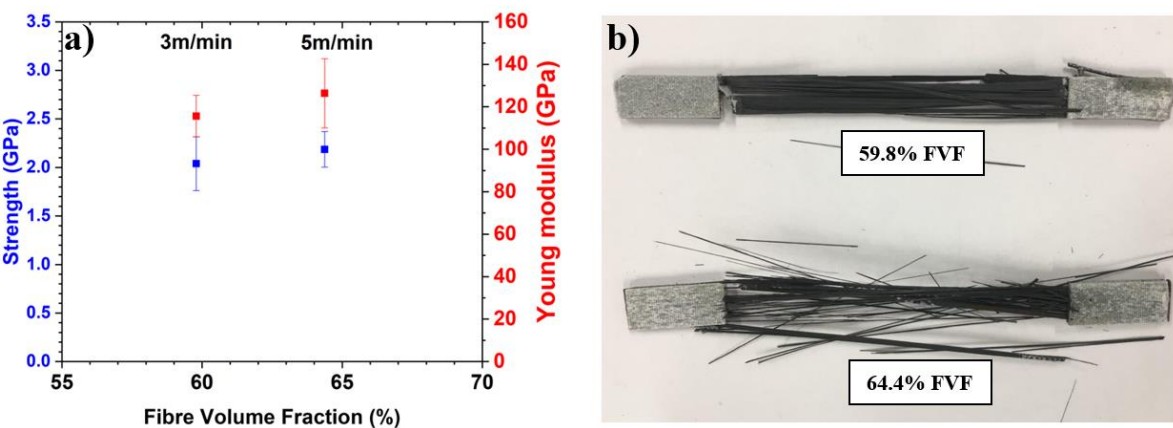

**Figure 9.** (**a**) Tensile test results of the UD-CFRP plates. (**b**) Different splitting modes of samples.

The Rule of Mixtures (ROM) is a simple method to estimate the mechanical properties of the composites, which can be defined for the longitudinal direction as [16]:

$$\sigma_c = \sigma_f FVF + \sigma_m (1 - FVF) \tag{7}$$

$$E_c = E_f FVF + E_m (1 - FVF) \tag{8}$$

where $\sigma_c$ is the tensile strength of the composite, $\sigma_f$ is the tensile strength of fibre, $\sigma_m$ is the tensile strength of matrix, $E_c$ is the modulus of the composite, $E_f$ is the modulus of the fibre, $E_m$ is the modulus, and *FVF* is the fibre volume fraction. Obtained FVF values from the experiments were used in Equations (7) and (8) to estimate the tensile strength and modulus of the composite plates. The discrepancy between the theoretical and experimental results stems from the fact that the ROM equation assumes perfect interface bonding of the fibre and matrix, which is not true in reality. As can be seen from the Table 3, the tensile test results deviated from the theoretical value by 30.73%, whereas the deviation for 64.4% FVF samples was 30.96%. As previously stated, fibre straightness has been found to have a considerable impact on tensile performance, with tensile strength dropping from 2650 to 1980 MPa (a 25% reduction) when the direction parameter *f* is reduced from 0.95 to 0.86 [14]. Given the calculated value of the samples' direction parameter (0.91), a divergence in tensile performance from optimal ROM values is to be expected. Lower values for tensile strength and stiffness can be attributed to the lower degree of fibre straightness of the samples.

**Table 3.** Comparison of measured tensile properties with theoretical values.

| Method | Tensile Strength at 59.8% FVF (MPa) | Modulus at 59.8% FVF (GPa) | Tensile Strength at 64.4% FVF (MPa) | Modulus at 64.4% FVF (GPa) | Fibre Direction Parameter *f* |
|---|---|---|---|---|---|
| Measured | 2039.04 | 115.63 | 2186.74 | 126.36 | 0.91 |
| ROM | 2913.67 | 138.75 | 3132.95 | 149.19 | 1 |

The adhesion between the fibres and the matrix depends on chemical bonding that can be improved by the addition of sizing agents to increase the bonding strength of intrinsically inert [22] and hydrophobic [23] carbon fibres, mechanical interlocking, or a combination of different factors [24]. Figure 9b shows standard fractured samples in tension for both 59.8% and 64.4% FVF. Higher FVFs mean lower resin content, and the interfaces between the fibre and the matrix may not be sufficient to provide an effective stress transfer. It was observed that samples with 64.4% FVF splintered in smaller bits than its lower FVF counterpart. The behaviour is consistent with a progressive fracture mode, which is caused by the inconsistent load distribution leading to premature failure of the weakest portions of the cross-section before spreading to the entire area. In contrast, samples with the lower FVF (59.8%) fractured in splitting mode and retain a better macroscopic consistency, which is compatible with an explosive failure profile, indicating a better interfacial bonding of the fibre and resin matrix.

### 3.5. Flexural Performance

Flexural properties are of great importance for many composite applications. For instance, tidal turbine blades endure extreme flapwise bending moments (up to six times larger than edgewise moments) because of the thrust loadings by the water [25]. For the same output, tidal turbine blades can be subjected to twice the thrust of wind turbines [26]. Due to the importance of flexural behaviour of composite structures, four-point bending tests were carried out, and the results are illustrated in Figure 10.

Interestingly, the flexural strength of the 64.4% FVF sample set (1.21 GPa) was 3.5% lower than its 59.8% FVF counterpart (1.26 GPa) while having 10.5% greater flexural modulus (115.04 GPa compared to 127.1 GPa). The higher interfacial porosity at very high FVF (see Figure 7) hastens the appearance of compressive buckling behaviour. Indeed, interfacial porosity created localised interfacial stress, resulting in early failure comparatively to less porous samples.

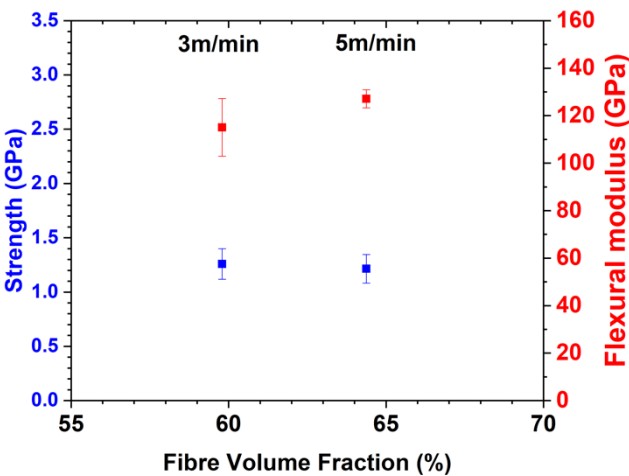

**Figure 10.** Flexural performance of the UD-CFRP plates.

### 3.6. Interlaminar Shear Strength (ILSS)

The ILSS test is a cohesive failure test usually used as a quality check in the composites industry due to the low amount of material required and the fast test speed [27]. With this method, material homogeneity, defects, and bulk porosity can be evaluated, especially if it occurs at the centreline of the specimen, where the shear stress is highest.

Figure 11 illustrates the change of ILSS of composite plates at different speeds and FVFs. Samples with 64.4% FVF display poorer interlaminar shear strength compared to the 59.8% FVF samples: 64.24 MPa compared to 58.49 MPa (an 8.95% decrease). The void content of 64.4% FVF samples is also higher than the 59.8% samples (32.2% higher void content). The authors believe that bulk porosity is the main contributing factor for the decrease in ILSS at higher production speed. Moreover, the much higher standard deviation (±6.24 MPa) of the 64.4% FVF sample set points to a less consistent consolidation in different samples due to the lack of resin to reach a high consolidation and therefore reduce bulk porosity.

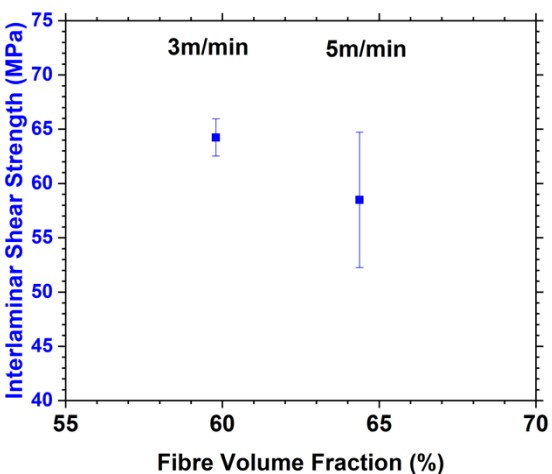

**Figure 11.** Interlaminar shear strength test results of the UD-CFRP plates.

### 3.7. Scanning Electron Microscope (SEM) Analysis

SEM micrographs of fractured samples were compared for both production speeds to visualise the fracture surfaces of the samples. The micrograph of the fractured 59.8% FVF sample (3 m/min) (Figure 12a) describes a homogeneous and brittle fracture: the interfaces show river lines, which are typical of cohesive failure and highlight a good fibre–matrix interfacial adhesion. A lateral fracture can be seen, highlighting a sudden stress release behaviour compatible with explosive failure.

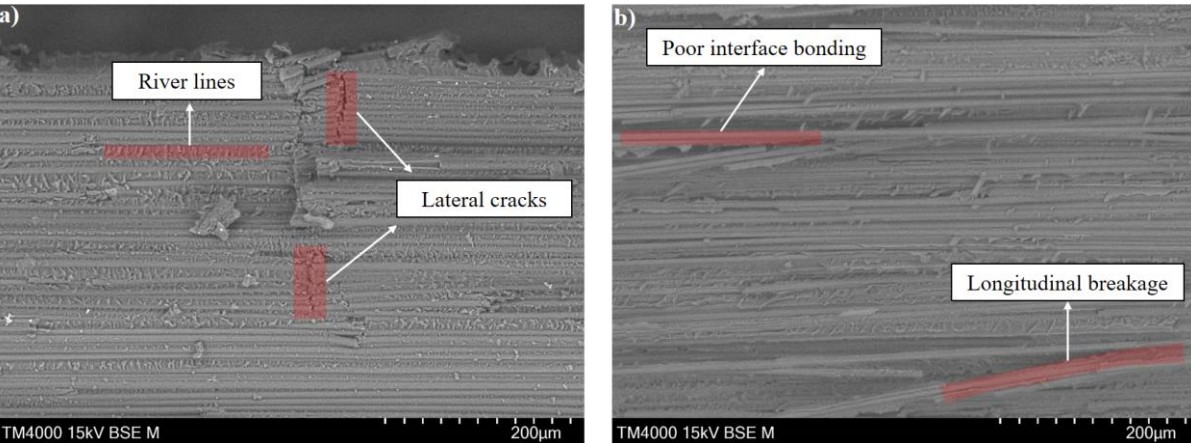

**Figure 12.** SEM images for UD-CFRP samples fractured in tension with (**a**) 3 m/min (59.8% FVF) (**b**) 5 m/min (64.4% FVF) production speed.

The higher FVF sample set (64.4%) highlights an entirely different behaviour. Clean interfaces can be seen, and the sample highlights longitudinal breakage (Figure 12b). These SEM images justify the conclusion drawn from the mechanical tests, as the lack of molten epoxy to fully coat individual fibre intra-tow regions caused a poorer interfacial stress transfer performance in general and allowed for interfacial localised stresses due to higher interfacial porosity.

## 4. Conclusions

The suitability of powder–epoxy in the high-speed, low-cost manufacturing of towpregs (prepreg tapes) has been demonstrated recently. These towpregs possess excellent properties for automated composite manufacturing systems such as AFP or filament winding. Controlling the fibre volume fraction (FVF), one of the most important parameters for the composite's performance, is crucial when high production speeds are targeted. With the reported tapeline system, it is possible to produce towpregs with high FVF and excellent mechanical properties; therefore, it offers tremendous potential as an alternative OOA approach. Furthermore, powder–epoxy offers low viscosity, virtually indefinite shelf life, the ability to co-cure multiple parts, and little to no VOC release. This study investigates the relationship between the production speed, FVF, and mechanical performance of the powder–epoxy-based carbon fibre towpregs. It was demonstrated that production at a slower speed (3 m/min) results in better consolidation, increased stress transfer, and better interlaminar adhesion due to the enhanced interfaces between the matrix and the carbon fibre reinforcement. It was observed that increasing the production speed may lead to inferior mechanical characteristics (up to 8.95% decrease in ILSS), although higher FVF values can be achieved. In addition, the interfacial void content of the samples was quantified by analysing optical micrographs with a custom algorithm. Results suggested that higher production speeds in the tapeline system results in higher interfacial voids (3.21% for 3 m/min and 6.07% for 5 m/min). A FVF of ≈60% for UD-CFRP laminates was found to be an optimal value, which can be reached with the proposed tapeline system by carefully controlling the production speed. Investigating the towpreg performance across a larger range and at higher production speeds (>10 m/min) could be of interest in establishing the optimal speed for the tapeline. Adjusting the powder particle flow to the tapeline speed in order to keep the FVF constant seems paramount in order to produce high-quality and homogeneous powder epoxy towpreg, and it is currently being investigated. Finally, a hand layup was shown to be problematic in regard to fibre alignment and resulted in lesser mechanical performances. Therefore, automation of the layup using an automated fibre placement (AFP) system is also of high interest for future investigations.

**Author Contributions:** Conceptualization, M.Ç., C.M.Ó.B., C.R.; Data curation, M.Ç., C.M.Ó.B., C.R., T.N., F.J., R.J.; Formal analysis, M.Ç., C.R., T.N.; Funding acquisition, C.M.Ó.B. and C.R.; Investigation, M.Ç., C.M.Ó.B., C.R., T.N., F.J., R.J.; Methodology M.Ç., C.R., T.N., F.J., R.J.; Project administration, C.R.; Resources, C.M.Ó.B. and C.R. Software, M.Ç. and T.N.; Supervision, M.Ç., C.M.Ó.B. and C.R.; Writing—original draft, M.Ç., T.N., F.J., R.J. and C.R. Writing—review & editing, M.Ç., C.M.Ó.B. and C.R. All authors have read and agreed to the published version of the manuscript.

**Funding:** This study was funded by CIMCOMP EPSRC Future Composites Manufacturing Research Hub (EP/P006701/1). The APC was kindly offered by Journal of Composite Science.

**Institutional Review Board Statement:** Not applicable.

**Informed Consent Statement:** Not applicable.

**Data Availability Statement:** Not applicable.

**Acknowledgments:** The authors would like to thank the CIMComp hub for financing this study, as well as Freilacke, Swiss CMT, and Toray C.A. for providing the needed materials for the investigations.

**Conflicts of Interest:** The authors declare no conflict of interest.

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
