# Peer review of "Influence of Line Processing Parameters on Properties of Carbon Fibre Epoxy Towpreg"

_jcs, doi:10.3390/jcs6030075_

Round 1

Reviewer 1 Report

General comments

The submitted manuscript presents the results on automation of the novel manufacturing process for carbon fibre powder-epoxy towpregs with maintained manufacturing properties. The authors presented an overview of the advantages and properties of powder-epoxy carbon composites as well as their own recent developments in the manufacturing of these materials. The impact of this study is an investigation of the automation of the manufacturing process, as well as a comparison of the resulting mechanical properties of composites manufactured with the proposed automated method with two selected manufacturing speeds. In section 2, the materials, manufacturing process as well as performed tests and processing algorithms, aimed at characterization of various properties of manufactured composites, were presented. In section 3, the authors presented the results of the evaluation of composites manufactured with two different manufacturing speeds using the developed automated line. The authors analyzed the post-production properties of fibre volume fractions and straightness, the presence of voids, compared the experimental results with the prediction using the rule of mixtures, as well as mechanical properties with a necessary justification and analysis of fracture mechanisms. The manuscript is original and interesting, presenting results on the novel automation approach of manufacturing composites using the OOA technique, and the presented results are valuable, primarily from the practical point of view. Some minor revisions are recommended before a publication.

Detailed comments

1) Please check the references in text, in some cases they are not well formatted.

2) Please comment on which basis the manufacturing speeds were selected in this study.

3) Please provide full information on a type of sensors, controllers, and other devices used in the production process, including their models and manufacturers.

4) All variables need to be written with italic font, please correct.

5) Since only two speeds of production were considered, it is suggested to add recommendations on the optimization of the production process with respect to production speed.

6) It would be beneficial to enrich conclusions with quantitative data obtained in experimental studies.

Author Response

Response to Reviewer 1:

1) Please check the references in text, in some cases they are not well formatted.

All the references were fixed, which were caused by conversion of the submitted manuscript to the template.

2) Please comment on which basis the manufacturing speeds were selected in this study.

This comment was addressed with the following sentence in the text at page 4, line 125-128: “Reliability and consistency were the goals for towpreg production, which are easier to achieve at slower production speeds.”

3) Please provide full information on a type of sensors, controllers, and other devices used in the production process, including their models and manufacturers.

Information on all the used equipment were added to page 3, line 98-103: “New sensors were added, including OS-PC30-2M-1V infrared temperature sensors (OMEGA Engineering), an SFD 500T tension sensor (Hans Schmidt & Co GmbH), and a P500 rotary sensor (Positek). Control hardware included a B35-HM magnetic particle brake (Placid Industries Inc), and EM-175 DC motor controllers. Automated tempera-ture and tension control was achieved using software-side PIDs on LabVIEW (National Instruments Corp.).”

4) All variables need to be written with italic font, please correct.

Variables were corrected, which were caused in the formatting process.

5) Since only two speeds of production were considered, it is suggested to add recommendations on the optimization of the production process with respect to production speed.

Recommendations on the future work regarding production speed was added to the conclusions part: “Investigating the towpreg performance across a larger range and at higher production speeds (>10 m/min) could be of interest in establishing the optimal speed for the tape-line.”

6) It would be beneficial to enrich conclusions with quantitative data obtained in experimental studies.

The authors would like to thank Reviewer 1 for his diligent reviewing and pertinent comments. Quantative data from the experimental work was added to the conclusions: “It was observed that increasing the production speed may lead to inferior me-chanical characteristics (up to 8.95% decrease in ILSS), although higher FVF values can be achieved. In addition, interfacial void content of the samples was quantified by an-alysing optical micrographs with a custom algorithm. Results suggested that higher production speeds in the tapeline system results in higher interfacial voids (3.21% for 3 m/min and 6.07% for 5 m/min).”

Reviewer 2 Report

The paper is well written and describe accuratelly the whole process, from the fiber impregnation to the laminates characterization. All the experiments are well designed and the results are clearly related to each other. I have only few correction to do:

1) verify the immages captions because there are reference errors with the figures;

2) i suggest to accurately check the presence of typing errors.

Author Response

Response to Reviewer 2:

  • verify the images captions because there are reference errors with the figures;

All the caption errors were fixed, which were originally caused due to the format editing.

  • I suggest to accurately check the presence of typing errors.

The manuscript was checked again for typing errors.

Reviewer 3 Report

This paper is well-written and organized. Therefore, I recommend publication. 

Author Response

Response to Reviewer 3:

This paper is well-written and organized. Therefore, I recommend publication. 

            The authors would like to thank Reviewer 3 for his comments.

Reviewer 4 Report

Authors have presented an interesting work on “Influence of line processing parameters on properties of carbon 2 fibre epoxy towpreg”, where fibre straightness and tensile properties are investigated. Work is well presented and is support with results. Below comments will be useful to improve the article further.

  • Page 3, Line 125: add the reference for equation (1). Similarly add references of other equations where possible.
  • Some of the references are missing, I believe it happened due to Latex error when compiling. See Page 3 onwards.
  • Presentation of Table 3 could be better. It is suggested to make table 3 similar in presentation to Table 1 and 2.

Author Response

Response to Reviewer 4:

  • Page 3, Line 125: add the reference for equation (1). Similarly add references of other equations where possible.

The authors would like to thank Reviewer 4 for his comments. References were added for the equations at page 4, line 130 and page 11, line 336, which is “P. Alam, Composites Engineering: An A–Z Guide. IOP Publishing, 2021.” The only other equation that required reference, which was for fibre straightness, already had the reference “D. Mamalis, T. Flanagan, and C. M. Ó Brádaigh, “Effect of fibre straightness and sizing in carbon fibre reinforced powder epoxy composites,” Compos. Part A Appl. Sci. Manuf., vol. 110, pp. 93–105, Jul. 2018.”

  • Some of the references are missing, I believe it happened due to Latex error when compiling. See Page 3 onwards.

The reference issues were fixed, which is due to the conversion process to the template and does not reflect the original manuscript.

  • Presentation of Table 3 could be better. It is suggested to make table 3 similar in presentation to Table 1 and 2.

The formatting of Table 3 was changed according to the Table 1 and 2.

Round 2

Reviewer 1 Report

The authors provided a detailed response to the comments from the first round of review and made appropriate corrections and extensions in the manuscript. I recommend this manuscript for a publication in its present form.

Reviewer 2 Report

Thanks to the authors that have carefully answered to my request

Reviewer 4 Report

Authors have incorporated the suggestions. Manuscript has been improved and publication is recommended.